# The Relationship between Emotion Regulation (ER) and Problematic Smartphone Use (PSU): A Systematic Review and Meta-Analyses

**DOI:** 10.3390/ijerph192315848

**Published:** 2022-11-28

**Authors:** Siti Hajar Shahidin, Marhani Midin, Hatta Sidi, Chia Lip Choy, Nik Ruzyanei Nik Jaafar, Hajar Mohd Salleh Sahimi, Nur Aishah Che Roos

**Affiliations:** 1Department of Psychiatry, Faculty of Medicine, Universiti Kebangsaan Malaysia Medical Centre, Kuala Lumpur 56000, Malaysia; 2Hospital Bahagia Ulu Kinta, Tanjung Rambutan 31250, Malaysia; 3Department of Psychiatry and Mental Health, Hospital Keningau, Peti Surat 11 Jalan Apin-Apin, Keningau 89007, Malaysia; 4Faculty of Medicine and Defence Health, National Defence University of Malaysia, Kuala Lumpur 57000, Malaysia

**Keywords:** emotion regulation, affect regulation, problematic smartphone use, smartphone addiction, mobile phone, meta-analyses

## Abstract

Emotion Dysregulation (ED) and Problematic Smartphone Use (PSU) are two rising global issues requiring further understanding on how they are linked. This paper aims to summarize the evidence pertaining to this relationship. Five databases were systematically searched for published literature from inception until 29 March 2021 using appropriate search strategies. Each study was screened for eligibility based on the set criteria, assessed for its quality and its level of evidence was determined. The Comprehensive Meta-Analysis software program (CMA) was employed to run further analyses of the data. Twenty-one studies were included in the systematic review. Nine studies with extractable data for meta-analysis had high across-studies heterogeneity, hence subgroup analyses were performed that confirmed a significant moderate positive correlation between ED and PSU (pooled correlation coefficient, r = 0.416 (four studies, *n* = 1462) and r = 0.42 (three studies, *n* = 899), respectively) and a weak positive correlation between “expressive suppression” and PSU (pooled correlation coefficient, r = 0.14 (two studies, *n* = 608)). Meta-regression analysis showed a stronger correlation between ED and PSU (R^2^ = 1.0, *p* = 0.0006) in the younger age group. Further studies to establish and explore the mechanisms that contribute towards the positive link between ED and PSU are required to guide in the planning of targeted interventions in addressing both issues.

## 1. Introduction

Emotion is a crucial determinant of behavior and its regulation facilitates adaptation, which is important for a good level of functioning and a sense of well-being. On the contrary, difficulties in emotion regulation (ER), termed emotion dysregulation, underlie a variety of psychopathology and contribute significantly to the development and maintenance of many psychiatric disorders [1]. ER is an evolving concept of the control of emotions through a range of responses that are tolerable and socially agreeable [2,3,4,5,6]. Gratz and Roemer [7] proposed that ER has multiple dimensions involving: (a) the awareness, understanding, and acceptance of emotions; (b) the utilization of adaptive skills to modify the intensity of emotional responses; and (c) the control of behavior, including repressing impulsive reactions and using goal-directed behavior during emotional distress. Gross and Thompson [8] further described ER as either automatic or effortful, conscious, or unconscious, internal, or external processes in attempts to be harmonious with oneself. ER may occur effectively along the emotion-generative course via two different focus strategies i.e., antecedent-focused, or response-focused, and is reflected through either cognitive reappraisal or expressive suppression [9]. Cognitive reappraisal is used as an initial strategy that works by altering the perception of the event causing the emotion and effectively modifies the resulting emotional response [10]. Cognitive appraisal is generally associated with more positive feelings and hence is said to play a role in enhancing subjective happiness [11] and a sense of well-being [12,13]. In contrast, expressive suppression changes the behavioral aspects of emotional response tendencies without processing the experience of emotions; therefore, does not influence either the expression of positive emotions or the reduction in negative emotions [14]. It is generally viewed as a maladaptive regulation strategy [11], may increase negative emotions and depressive symptoms [15], and reduce subjective happiness. Understanding these mechanisms provides valuable insight into the process of ER and its relationship with psychological well-being.

ER was broadly conceptualized in multiple domains with subsequent development of multiple instruments for its measurement to capture the whole scope. [16]. Gross and John [11] developed their instrument, called the Emotion Regulation Questionnaire (ERQ), designed to assess the habitual use of cognitive reappraisal and expressive suppression that differs between individuals. Additionally, Gratz and Roemer [7] invented the Difficulties in Emotion Regulation Scale (DERS), which measures trait-level perceived emotion regulation ability as per their definition, with higher total scores reflecting a higher degree of emotion regulation difficulties. It consists of six subscales: (a) lack of emotional awareness (AWARENESS), (b) lack of emotional clarity (CLARITY), difficulty regulating behavior when distressed (IMPULSE), difficulty engaging in goal-directed cognition and behavior when distressed (GOALS), (e) unwillingness to accept certain emotional responses (NON-ACCEPTANCE), and (f) lack of access to strategies for feeling better when distressed (STRATEGIES). The Cognitive Emotion Regulation Questionnaire (CERQ) is another questionnaire developed to assess cognitive components of emotion regulation [17]. This questionnaire assesses nine cognitive emotion regulation strategies, (i.e., self-blame, other-blame, rumination or focus on thought, catastrophizing, putting into perspective, positive refocusing, positive reappraisal, acceptance, and refocus on planning) by gauging an individual’s thought response after experiencing a stressful threat or events. The few examples above briefly illustrate that each of the instruments measures a different aspect of ER according to how the term was approached. Nonetheless, they also have some overlapping features, which in this case are worth analyzing in this study.

When emotion becomes dysregulated, attempts to attenuate the resulting emotional intensity could include maladaptive coping in the form of behavioral addiction; among which is problematic smartphone use (PSU). Smartphones are a highly addictive technology that enveloped society due to their ease of use, convenience, and almost infinite access. Furthermore, the increasing importance of smartphones in our daily life in these modern days blurs the line in determining whether individuals’ usage of smartphones is adaptive or maladaptive to their emotional coping. Smartphone use is a potentially maladaptive mechanism to ER as its use exposes a person to various emotionally triggering stimuli, which could further dysregulate one’s emotions.

PSU is defined as the inability to regulate one’s use of the mobile phone, resulting in negative consequences in the daily life of its user [18]. Its estimated prevalence is up to 38% depending on the definition, setting, and scales used to quantify the behavior [19,20]. Younger people, as the most frequent users of smartphones and the internet, are more likely to be affected by problematic use. A recent systematic review and meta-analysis [21] reported that as high as approximately one in every four children and young people (CYP) had PSU. Problematic smartphone or mobile phone use (PSU) is generally considered maladaptive in coping with stress and negative emotions [22]. PSU is associated with stress, depression, and anxiety symptoms [23,24,25], loneliness [26,27,28], family conflict [26], sleep problems [29,30,31], low social support [32], lower academic performance in students [33,34], as well as perceived academic disturbance in adolescents [26]. Physically, PSU is linked to bodily discomfort in the head, neck, and eye [26,35,36].

In measuring PSU, the most efficient or gold standard scales are yet to be established; disagreements exist on how to approach this problem [37]. Existing literature tends to measure this construct in the context of the “addiction” framework following their conceptual similarity, thus the term “Smartphone addiction” is commonly seen to be used interchangeably with PSU [38]. Smartphone usage is also gauged in the background of associated motivations, attitudes, and frequency, which may reflect the mechanism behind PSU [39,40,41]. A relatively new term that sometimes is used interchangeably with PSU is nomophobia, derived from the word “no mobile phone phobia”, which refers to the anxiety and intense discomfort of not being in touch with a mobile phone and its associated features [42,43].

Due to the various psychopathology and mental health conditions that both PSU and ER are associated with, it would be important to understand the link between them. There are growing studies examining the relationship between ER and PSU globally with the consideration that these two complex phenomena are possibly linked in a non-linear manner. To the best of our knowledge, no systematic review and meta-analysis was conducted to weigh the evidence so far. Hence, this paper aims to examine this inter-relationship and provide an updated and comprehensive review of the role of emotion regulation in PSU and vice versa. A deeper understanding of their relationship is needed to identify the next targets and opportunities for intervention, particularly to enhance ER and identify strategies to curb PSU.

## 2. Methods

This review is in accordance with the Preferred Reporting Items for Systematic Reviews and Meta-Analyses (PRISMA) guideline [44,45] in the design, workflow, and reporting of the obtained results. It is a guideline recommendation consisting of a 27-item checklist to identify key features, with a four-phase flow diagram essential for transparent reporting of a systematic review. This systematic review was registered in the International Prospective Register of Systematic Reviews (PROSPERO) and published on the website on 25 April 2021, with the registration number CRD42021244575 [46].

### 2.1. Search Strategy

Five electronic databases, i.e., PubMed, Ovid Medline, Cochrane Library, Web of Science, and Scopus, were systematically searched for published literature. Materials that existed from inception until 29 March 2021 were gathered by applying their own developed search strategy established from keywords with relevant truncation (marked with asterisks) and Boolean operators of AND and OR across the selected electronic databases. For Emotion Regulation (ER), ensuing keywords were used: “emotion* *regulation”, “mood *regulation”, “affect* *regulation”, “emotion*”, and “labil*”. Meanwhile, for PSU, keywords used were “problematic smartphone use”, “problematic use”, “addict*”, “maladaptive”, “smartphone”, “hand phone”, “mobile phone”, and “cell phone”. The search strategy was limited to studies enrolling human subjects and articles published in English. No geographical restriction was applied. To increase the chances of identifying relevant studies, a bibliographic search was carried out at the beginning of the research from related systematic reviews and eligible studies.

### 2.2. Eligibility Criteria

Studies included in the review met the following criteria: (1) primary quantitative studies of observational/longitudinal, cross-sectional, or cohort study design; (2) included smartphone or mobile users of all ages and genders from a healthy general population; (3) measuring levels, components, or dimensions of ER to PSU and vice versa; and (4) full papers published in the English language. The exclusion criteria set were (1) articles published in non-English language; (2) case reports, reviews, unpublished studies, conference abstracts, trial protocols, and proceedings; and (3) special population studies, meaning studies including samples/subjects having pre-existing DSM 5 or medical diagnoses.

### 2.3. Selection Process

Search results obtained were all exported to reference management software (Covidence systematic review software, Veritas Health Innovation, Melbourne, Australia). This software automatically sorts out duplicate studies upon import, enabling easier screening of titles and abstracts by two independent reviewers. Next, the full texts were carefully determined for eligibility based on the inclusion and exclusion criteria set earlier. Any discrepancies between the two reviewers (SH and other team members) were resolved by a third reviewer or via group consensus

### 2.4. Data Extraction and Management

Two independent reviewers (SH and LC) performed the data extraction into Microsoft Excel spreadsheets (Microsoft Corporation, Redmond, WA, USA). Any conflicts with the extracted data were resolved by a third reviewer (MM) or group consensus. Items extracted included information concerning descriptive data and statistical data for running the meta-analysis. The descriptive data extracted include (1) first author and publication year, (2) country of origin, (3) study setting, (4) sample characteristics (sample size, age group), (5) PSU measures and domains, (6) ER measures and domains, and (7) the descriptive relationship between PSU’s and ER’s domain with *p*-value expressed in *. The statistical data for running the meta-analysis extracted include (1) PSU measures (mean, SD), (2) ER measures (mean, SD), (3) sample size, (4) correlational coefficient together with its 95% Confidence Interval (CI), (5) *p*-value, and (6) the relevant data for running meta-regression. Relevant authors were contacted [47,48,49,50,51,52,53] for clarification of incomplete data, in which all but 1 author replied to get extra information required for the review and analysis.

### 2.5. Quality of Study

A modified version of McMasters Critical Appraisal Tool for quantitative studies checklist was utilized to determine the quality of each included study, whereby the domains assessing interventions were taken out to suit the etiological nature of the studies included in this review. A rating of “yes”, “no”, “not addressed”, or “not applicable” was assigned to each question with a 1-point score given to “yes” answers. This brings the highest possible total score of 11, and variation may arise by excluding questions rated “not applicable” in keeping with different study designs. The level of evidence of the included studies adhered to the Australian National Health and Medical Research Council’s (NHMRC) evidence hierarchy [54]. The five components assessed were: (i) evidence base, (ii) consistency of findings across included studies, (iii) clinical impact, (iv) generalizability, and (v) applicability. Each component was given a grade from “A” to “D” to help guide the weighing of the recommendation. These two processes were assessed by two independent reviewers (SH and LC) and any conflicts were resolved by a third reviewer, which could be anyone else from the group.

### 2.6. Statistical Analysis

A meta-analysis was performed using the Comprehensive Meta-Analysis software program (CMA) (Biostat Inc., Englewood, NJ, USA) to determine the relationship between emotional regulation with PSU. Only studies reporting the Pearson’s correlation coefficient between ER and PSU were included in the meta-analysis. The Pearson’s correlation coefficient together with its 95% CI was computed into CMA where the pooled effect estimates were automatically generated by the meta-analysis software. A random effect (RE) model was used for the summarized effect estimates to account for heterogeneity across studies. Analysis of data was conducted according to the different scales used to assess ER and PSU, e.g., DERS, EMSS, and ERQ scale for ER versus SAS, SPAI, TMD, and APU scale for PSU. The I^2^ index statistic was used to describe the heterogeneity between the included studies. An I^2^ value of 0% implies no observed heterogeneity, while greater values indicate higher heterogeneity [55]. Meta-regression was conducted to explore factors such as age and gender that may explain the heterogeneity observed. A result with a *p*-value of <0.05, was deemed to be statistically significant. A sensitivity analysis was conducted by the exclusion of the study or group of data for the evaluation of the result’s robustness. A publication bias analysis was not reported as the meta-analyses consisted of less than ten studies.

## 3. Results

### 3.1. Systematic Review

#### 3.1.1. Study Selection

A sum of 2774 studies was obtained via online database searches, with 544 duplicate records. There were 2106 studies from Ovid Medline, 300 from Web of Science, 217 from Scopus, 95 from PubMed, 53 from Cochrane Library, and 3 from preliminary literature searches through the reference list. Five-hundred-forty-four studies were excluded post-removal of duplicates, and a further 2129 records were excluded after the screening of titles and abstracts. A total of 101 studies with full-text manuscripts were examined for eligibility. Figure 1 shows the rationale for excluding each study and the whole selection process. Finally, 21 studies were included in the systematic review.

#### 3.1.2. Quality Assessment

In view of all the included studies being cross-sectional in design, they were rated as level IV, as per the NHMRC evidence hierarchy used to determine the level of evidence in each study. In assessing the risk of bias, a modified McMaster Critical Appraisal Tool for quantitative studies was employed and measured in terms of percentage according to the number of criteria fulfilled. As shown in Appendix A, most studies have relatively good qualities, whereby 1 study scored 100% [40], 6 studies 91% [49,50,51,56,57,58], 12 studies scored 82% [47,48,52,59,60,61,62,63,64,65,66,67], and another 2 studies score 73% [53,68] (refer to Appendix A).

#### 3.1.3. Study Characteristics

The majority of the studies (90.4%) were recently published in the last five years (2017–2021) and carried out in nine countries, namely the USA, Italy, Turkey, China, Hungary, Canada, Australia, Spain, and Brazil. There were a total of 8223 subjects altogether, with mean ages ranging from 13 to 48 years across studies. All studies applied convenience sampling methods with large portions of the samples collected from school and university settings. Only one study involved multiple informants in the form of triads of parents and children, thereafter, a non-youth population was included in their samples [49]. The included studies managed to capture subjects of variable age groups, from those in their teens, 20s, 30s, and 40s. Italy was the biggest contributor to the collected data, having provided a total number of 1925 subjects altogether (Table A1 of Appendix B).

#### 3.1.4. Emotion Regulation Measures

The studies employed the usage of designations such as “emotional dysregulation”, “emotional control difficulty”, “dysfunctional emotion regulation”, and “expressive suppression” to refer to maladaptive ER or ED, whilst the term “cognitive reappraisal” is used to refer to adaptive ER. Nine of the studies used DERS [48,49,50,51,53,58,59,61,65], seven studies used ERQ [40,47,52,56,57,64,68], one study [56] other than using ERQ, also used an additional scale, which is the Ruminative Thought Style Questionnaire (RTSQ), two studies used CERQ [63,67], one study used the Mood Regulation Scale (MR) [60], one study used EMSS [62], and one study used ERSA [66]. All scales are validated.

#### 3.1.5. Problematic Smartphone Use Measures

Terms such as “problematic smartphone use”, “smartphone addiction”, “smartphone overuse”, “smartphone amount use”, “smartphone use frequency”, and “nomophobia” were summarized according to findings as “problematic smartphone use”, abbreviated as “PSU”. Twelve studies used SAS (among them, seven studies used the short version) [40,47,48,51,52,53,56,57,63,65,66,67], Elhai et al. [57] and Elhai and Contractor [56], who used SAS, also used an additional validated smartphone usage scale developed by the author, three studies used SPAI [49,50,64], in which Fortes et al. [64] also used the Habitual Smartphone Use Questionnaire other than SPAI, one study used the Nomophobia Questionnaire (NMP-Q) [62], one study used the Adolescent Preoccupation with Screen Scale adapted to adult smartphone use [58], one study used TMD-brief [59], one study used the Smartphone Addiction Scale (SA) [60], one study used the Addictive Patterns of Use Scale (APU) [61], and one study used the Mobile Phone Use Scale [68]. All scales are validated except for the Habitual Smartphone Use Questionnaire, which was developed exclusively for the study by Fortes et al. [64].

#### 3.1.6. Relationship between ER and PSU

Table A2 and Table A3 in the Appendix B summarize the key findings from the studies on this relationship. In studies using PSU total scores in measuring PSU, 17 studies reported a significant positive correlation between ED and PSU, with only 1 study [62] reporting no significant correlation.

Another three studies measured different domains of PSU instead of PSU total scores and yielded specific relationships between the PSU domains and ED. Elhai and Contractor [56] reported a significant positive correlation of ED with the “positive anticipation”, “cyberspace-oriented relationship”, and “tolerance” domains of SAS, while no correlation was found in the “daily life disturbance” and “withdrawal” domains of SAS, and “frequency of use” measured by a validated self-developed scale. Fortes et al. [64] reported a significant positive correlation of ED with the “functional impairment” dimension of the Smartphone Addiction Inventory (SPAI) and no correlation with the “amount of use”. Interestingly, Hoffner and Lee [68] reported a positive correlation between “expressive suppression” with PSU only in the content of entertainment or information, but not PSU on interpersonal contact and social support.

#### 3.1.7. Relationship between Specific ED Domains and PSU

Among domains of ED, we found a consistent statistically significant positive correlation between seven domains of ED and PSU, i.e., “expressive suppression”, “impulse control difficulties”, “lack of emotional clarity”, “rumination”, “catastrophizing”, “self-blame”, and “blaming others”. In seven studies reporting on “expressive suppression”, five studies [40,47,52,56,68] reported a positive correlation between “expressive suppression” and PSU. One study [57] only found a positive correlation between “expressive suppression” with “baseline objective minutes of smartphone use” and also the “trend” of “objective minutes of smartphone use per weekday”, but no correlation with the “objective self-reported smartphone uses per day or weekday or weekend. Although this study used the total SAS score, the author only reported a significant positive correlation between the SAS score with objective measures of smartphone use, without directly reporting the correlation of the SAS score with “expressive suppression”. Among the seven studies reporting “expressive suppression”, only one study [64] reported no correlation between “expressive suppression” and PSU.

Among the nine studies that employed the DERS scale, only two studies [58,59] provided details on the six domains of ED. These two studies revealed consistent significant positive correlations of two domains, i.e., “impulse control difficulties” and “lack of emotional clarity” with PSU. Both studies reported no correlation between “lack of emotional awareness” with PSU. Two studies using CERQ [63,67] and one study using RTSQ [56] consistently found that “rumination” significantly and positively correlated with PSU. Additionally, two studies using CERQ [63,67] consistently reported significant positive correlations of three domains, i.e., “catastrophizing”, “self-blame”, and “blaming others” with PSU separately.

#### 3.1.8. Relationship between Cognitive Reappraisal of ER and PSU Domains

Six studies assessing the link between “cognitive reappraisal”, i.e., the adaptive form of ER and PSU, revealed inconsistent findings. No correlation between “cognitive reappraisal” and PSU was found in three studies [40,57,64], while a significant positive correlation was found in two studies [56,68], and a significant negative correlation was found in one study [47]. The two studies reporting on the positive correlation between “cognitive reappraisal” and PSU domains reported a correlation with different domains of PSU. Elhai and Contractor [56] reported a significant positive correlation between “cognitive reappraisal” with PSU domains of ‘smartphone use frequency’, the PSU ‘overuse’, ‘positive anticipation’, and ‘tolerance’ subdomains, but not on ‘daily life disturbances’, ‘cyberspace-oriented relationship’, and ‘withdrawal’ subdomains. Hoffner and Lee [68] found a significant positive correlation of “cognitive reappraisal” with PSU on the content of interpersonal contact and social support, but not on entertainment or information content.

### 3.2. Meta-Analyses

Our systematic review and meta-analyses found a relationship between ER and PSU. Nine studies provided extractable data for meta-analysis to calculate the correlation of ED with PSU (*n* = 3793). Among them, Giordano et al. [49] provided three and lo Coco et al. [50] provided two data from different groups of subjects; therefore, 12 sets of data were entered for meta-analysis. We set the program to run with a random effect model in view of different age groups, scales, and domains of ED, as well as different scales used in defining the PSU. We found a pool correlation coefficient r = 0.286 (*n* = 3793, 95%CI 0.160–0.402, *p* < 0.0001, I^2^ = 94.03%); however, because of high heterogeneity across studies, it is deemed not appropriate to apply a meta-analysis of correlation [69,70]. We furnished the relevant data of this analysis only in Appendix A and proceeded to subgroup analysis.

### Subgroup Analysis

In view of high I^2^, we ran subgroup analysis in studies using the same scales measuring the same domains.

DERS and SAS scales

There are four studies using DERS to measure difficulties in ER and SAS to measure PSU as shown in Figure 2a,b. Based on Figure 2a,b, we found a pool correlation coefficient, r = 0.416 (*n* = 1462, 95% 0.372 to 0.457, *p* < 0.001, I^2^ = 0%), indicating a stronger positive correlation between ED and PSU.

2.DERS-SF and SPAI scales

Figure 3a,b shows two studies using DERS-SF and SPAI scales, with a total of five sets of data. We found a pool correlation coefficient, r = 0.356 (95% CI 0.272 to 0.434, *p* < 0.001, I^2^ = 67.72%). A meta-regression analysis was run to investigate factors that may explain the heterogeneity, as we noted the I^2^ was high in this subgroup analysis and found that age is the factor that explains it. Meta-regression analysis shows that the younger the subjects’ age, the stronger the correlation between emotional regulation and PSU (R^2^ = 1.0, *p* = 0.0006), while gender has no significant impact on the correlation (Figure 4). Subsequently, we narrowed down the subgroup analysis by removing the two parents’ groups. This time we found a pool correlation coefficient, r = 0.42 (*n* = 899, 95% CI 0.364 to 0.472, *p* < 0.001, I^2^ = 0%) (Figure 5a,b).

3.ERQ and SAS

There are two studies using ERQ and SAS exploring the domain of “expressive suppression”, as shown in Figure 6a,b. We found a highly homogenous finding, although a small positive correlation with pool correlation coefficient, r = 0.143 (*n* = 608, 95% CI 0.0639 to 0.220, *p* < 0.001, I^2^ = 0%), between “expressive suppression” and PSU.

### 3.3. Conclusion of Meta-Analysis

Overall, we obtained a pooled correlation coefficient of around 0.4 (r = 0.416 (four studies, *n* = 1462) and 0.42 (three studies, *n* = 899), respectively) in subgroup analysis. Thus, we confirmed a significant moderate positive correlation between ED (measured by DERS and DERS-SF) with PSU (measured by SAS and SPAI). We also found a significant, but weakly positive, correlation of expressive suppression with PSU, with a pooled correlation coefficient of 0.14.

### 3.4. NHRMC Evidence Statement Matrix

It is important for individuals to be aware of the possible hazardous use of smartphones in relation to their day-to-day emotional state. This systematic review highlights a weak to moderate positive correlation between ED and PSU. The NHRMC evidence statement matrix is presented in Table 1. The findings indicate some reasonable evidence to support the association, and hence an overall grade C recommendation was given.

## 4. Discussion

This systematic review and meta-analysis determined the relationship between ER and PSU. A total of 21 studies met the inclusion criteria, involving a total of 8223 subjects. Overall, the findings indicate a significant correlation between ER and PSU. A systematic review of studies looking at associations between ED and total PSU scores in 17 out of 18 studies showed a significant positive correlation between emotional dysregulation and PSU. This is confirmed by a meta-analysis that showed a significant moderate positive correlation between ED (measured by DERS and DERS-SF) with PSU (measured by SAS and SPAI). Meta-regression analysis showed that the younger the subjects’ age, the stronger the correlation between ED and PSU, while gender seemed to not have any significant role in this association. 

Additionally, different domains of ED were found to be associated with different domains of PSU. Seven domains of ED were found to be consistently positively correlated with PSU, i.e., “expressive suppression”, “impulse control difficulties”, “lack of emotional clarity”, “rumination”, “catastrophizing”, “self-blame”, and “blaming others”. “Expressive suppression” was particularly confirmed through meta-analysis to be significant even though weakly correlated with PSU. However, “lack of emotional awareness” had no correlation with PSU in both studies, which provided these details. 

Studies reporting an association between ED and domains of PSU reported a significant positive correlation between ED and “positive anticipation”, “cyberspace-oriented relationship”, and “tolerance” domains of PSU, when measured by using SAS, and with “functional impairment” dimension of PSU when measured by using SPAI. Additionally, no correlation was found in the “daily life disturbance” and “withdrawal” domains of PSU. Another interesting finding was that “expressive suppression” was positively correlated with PSU only in the “content of entertainment or information”, but not in the “interpersonal contact and social support” domains of PSU. These findings are rather mixed and difficult to be interpreted and concluded in view of each finding being provided by a single study, and there are no repeated results. Nonetheless interestingly, “frequency of use”, measured by a validated self-developed scale, and “amount of use“, measured by using SAS, was not significantly correlated with ED. Two studies [56,64] included in this systematic review reported that frequency and amount of smartphones were not linked to ED, while Elhai et al. [57] only managed to find a positive correlation in this relationship on the weekdays throughout the 1-week duration of the study. Intriguingly, Elhai and Contractor [56] also reported a significant positive correlation between “cognitive reappraisal” with PSU domains of ‘smartphone use frequency’. These findings are rather difficult to explain. The context of the high frequency and duration of smartphone use was not clearly addressed in the studies, such as whether they are being used productively or maladaptively. Relevant to the sample characteristic of these studies of high school and university students, group work assignments or meetings, for example, may take up a lot of time, but such utilization clearly does not signal the pattern of PSU, nor point to any form of ER strategies, either adaptive or maladaptive. It takes more than just the amount or frequency of smartphone use to identify the user to be problematic.

With regards to our main finding, i.e., ED having a clear association with PSU, it is not surprising. Poor ER was predictive of all addictive behaviors [71]. PSU was mostly addressed as a form of behavioral addiction and is likened to a gambling disorder, with some striking similarities with substance abuse [72]. Those with poor or limited ER skills may engage in behaviors that can help extend and prolong positive emotional states [73]. Psychological relief obtained from giving in to smartphone cravings may perpetuate its use as well. ED is found to underlie various psychopathologies and mental health conditions, which are proven to be associated with PSU. These include depression, stress, anxiety disorders, and Fear Of Missing Out (FOMO) which proved to be related to both PSU and maladaptive emotion regulation [24,51,74]. As emotional suppression was established in this meta-analysis to have substantial linkage to PSU, the possibility of it being one of the mediums or outlets of choice to regulate emotions is highly likely, especially when a smartphone is equated as an extension of oneself [75]. The internet is an essential resource that permits the majority of the sought smartphone’s function. Therefore, the Interaction of Person–Affect–Cognition–Execution (I-PACE) process model of internet use disorders [76], is transferable to smartphone use as well. Interestingly, one study in this systematic review shows that when individuals utilize the smartphone to self-regulate by way of “emotional suppression”, they tend to go for process use of the smartphone by accessing the content or entertainment aspect of the device rather than the social use of utilizing its social interaction features [68], indicating motive of smartphone use as an important element in the relationship between ER and PSU. It is known that motive strongly drives PSU. The User Gratification Theory (UGT) motions that the prospect of getting specific gratifications drives the choice of certain media chosen by individuals [77]. Similarly, avoidance of interpersonal communication also may facilitate emotional suppression, as was demonstrated by a certain population group, such as those with social anxiety, who have a higher tendency to suppress their own emotions with a higher severity of the condition [68,78]. These different choices of smartphone activities may operate by providing distraction or escapism among other mechanisms, as suggested by the Compensatory Internet Use Theory (CIUT) [79].

Our meta-regression analysis showed that the younger the age group, the stronger the correlation between ED and PSU. Studies show that the younger age group has less use of acceptance, and more use of maladaptive strategies during intense emotional situations [80]. This could be explained by the development of neuro-circuity in the brain during adolescence, which leads to the development of emotional regulation. Casey et al. [81] postulated that hierarchical changes in the brain circuitry (subcortico-subcortical to subcortico-cortical to cortico-subcortical and lastly cortico-cortical) may account for the reason for the maturity of emotional regulation as we age. Older individuals may, over the years, develop other positive coping skills, established their identity, and have more access to supportive resources. In the younger population, parental variables such as lower education and income, younger parents, dual income families, i.e., both working parents, permissive parenting style, and positive parental attitudes towards smartphones, as well as parental heavy smartphone use [82], also contributed to a higher occurrence of smartphone addiction. The tendencies of the younger age group with ED and PSU are a big concern. This may explain the rise of mental health problems around the world in this age group. According to Malaysian data [83], suicidal behavior among teens increased, highest among 13-year-old students, with the statistics in 2017 being 11.2 percent for suicidal ideation, 10.1 percent for suicidal attempts, and 9 percent for suicidal plans. Suitable interventions are needed to promote well-being and help prevent young people from having this condition. Another result highlighted by the meta-regression analysis is that gender has no significant impact on the correlation between emotion dysregulation and PSU. The gender finding can be explained by the complexity of interactions of gender with individual characteristics, interpersonal contact, the environment, or the type of task in which they are observed [84,85]. This is complementary to the knowledge that gender differences in emotional regulation are dynamic, instead of a static trait [85]. Age, on top of being a significant strong influence on emotion regulation itself, appears to modulate the effect of gender on emotion regulation too.

Our study showed that among the different domains of ED, lack of emotional clarity, impulse control difficulties, and negative cognitive appraisal, such as catastrophizing, self-blaming, and blaming others, has a positive correlation with PSU. Emotional clarity, which is the individual’s ability to clearly identify and recognize his/her own emotions, is extremely important in emotion regulation [86]. Individuals with greater emotional clarity are able to pause and acknowledge their feelings when they are in emotional distress. They tend to use positive forms of coping mechanisms instead of using their smartphones in a negative way. Individuals with impulse control problems have difficulty controlling certain emotions or behaviors. Impulse control is linked with behavioral addiction, [87,88] hence, this positive correlation with PSU is reasonable. This link was explained via two interacting neural systems, which are the impulsive (amygdala dependent) system and the reflective (orbito-frontal dependent) system in the brain [89]. Meanwhile, negative cognitive appraisal, such as catastrophising, self-blaming, and blaming others (measured in CERQ) was found to significantly correlate with PSU. This shows the importance of looking into the cognitive re-evaluation strategies, such as the cognitive behavioral therapy approach, in the possible management of treating PSU. Lack of emotional awareness, however, was found to have no significant correlation with PSU. This domain of emotion regulation difficulties is extracted from one of the DERS subscales. The “Awareness” subscale alone showed relatively poor psychometric properties both in its original and short forms with poor internal consistency (α < 0.80), and convergent validity for the total scores was improved upon the exclusion of this item [90]. Conscious awareness of emotions is needed for “cognitive reappraisal” to be employed as an adaptive emotion regulation strategy [91]. Therefore, it can be concluded that emotional awareness may be a requirement, but an inadequate criterion for emotion regulation and does not seem to be the same construct. It could be possible that the “lack of awareness construct” measured by DERS was not fully captured by the model of levels of emotional awareness by Subic-Wrana et al. [91] which includes implicit and explicit emotional processing. 

This systematic review identified rumination to have significant positive correlations with PSU. Three of the included studies consistently reported this positive correlation. Rumination is depicted as an emotional process characterized by unsought, repetitive, past-oriented, and negatively inclined thoughts [92,93]. Rumination was related to worry, and therefore is regarded to be a maladaptive strategy that was proven to aggravate and sustain an array of mental health issues, such as depression [94,95], and to an extent to PSU. Rumination, similar to “expressive suppression”, has a positive relationship with psychological distress, whereby the level of distress is amplified when individuals engage in some form of rumination, contributing to feelings of being overwhelmed [96].

Our findings on the “cognitive reappraisal” and “expressive suppression” domains of ER and PSU, are worthy of discussion. While there is a consistent positive correlation between “expressive suppression” and PSU, the relationship between “cognitive reappraisal” is inconsistent. Cognitive reappraisal and expressive suppression are emotional regulation strategies measured by ERQ: cognitive reappraisal is changing the way one thinks about the potentially emotionally provoking situation, while expressive suppression is altering the way one behaviorally responds to those situations. “Expressive suppression” is considered a maladaptive form of emotional regulation, considering the negative outcomes on emotional and social functioning [97], making it part of ED. The inconsistent findings on the association between cognitive reappraisal and PSU may be explained by this construct having both negative and positive effects on the emotional experience [98]. It was postulated that cognitive reframing is not easy to achieve immediately when in a distressing situation [99] and hence, individuals have varying baseline capacities to perform cognitive reappraisal to begin with. This could explain the inconsistency in our result. We realized that the measurement for emotional regulation (DERS and ERQ) is testing different domains, albeit some are overlapping. ERQ test strategies are employed during highly stressful situations; the higher the score, the better the usage of those strategies. Meanwhile, DERS measures the six facets of difficulties in emotional regulation; the higher the scores, the higher the emotional dysregulation. This explains the heterogeneity in our results when we analyzed them together. Emotional regulation is a multidimensional construct that includes difficulties in regulating emotions and skills or strategies in regulating emotions. ERQ in itself is not enough to measure emotional dysregulation while DERS was shown to be superior to ERQ, as it was able to capture ER in the clinical context, as ERQ showed a weak association with psychiatric symptoms (depression and anxiety) compared to the strong association with DERS domains [100]. Despite a surge in research on emotional regulation topics, the full construct of emotional regulation is still far from being completely understood. The authors suggested that more future research on understanding the multi-dimensional construct of emotional regulation is needed and a more comprehensive measurement tool for emotional regulation that tests all domains should be explored.

Our systematic review and meta-analysis were unable to establish the direction of the relationship between ER and PSU, as all the studies were cross-sectional in nature. The study faced the challenges related to both ER and PSU being evolving concepts with different measurements capturing different aspects of both conditions. In a very recent 4-year longitudinal study looking at the causal relationship between compulsive internet use (CIU) and emotion regulation difficulties, it was found that CIU preceded the development of some aspects of ED, with no proof that emotion regulation difficulties preceded the development of increases in CIU [101]. Due to CIU carrying some degree of similarity with PSU, there is a likelihood that such results may replicate in the case of ER and PSU. The implication of this result changes the way future studies with regards to PSU need to be designed, as the available studies tend to address the possibility of dysfunction in ER as causing PSU and not the other way around. In this case, that would mean a direct approach to limiting smartphone use could potentially be more effective or suitable in comparison to focusing on emotion regulation strategies in preventing and managing PSU. Hence, there is a strong need for further studies to look at the direction of the relationship.

### Trengths and Limitations

This study has several limitations. ER is an evolving and wide concept, with many perspectives or views, many of them overlapping, making standardization of the findings difficult. Differences in measurement, which sometimes capture different aspects of ER, resulted in the heterogeneity of results when we compiled them all together. A single measurement alone is not suitable or enough to capture all aspects of ER. However, we still managed to run a few meta-analyses, as some studies still maintain some degree of homogeneity.

Meanwhile, although there is a blooming interest in the concept of PSU, it seems that this topic is still in its infancy, considering that smartphones only began being gradually used universally less than two decades ago. Even its status as a genuine behavioral addiction receives much debate from the scientific community. The commonly used definition by Billieux [18] implied that PSU also covers the non-psychological aspect, such as physical health, making it initially difficult to establish the focus of the discussion.

A number of challenges were faced in conducting the study, and we have taken several measures to solve them, for example, using consistent studies utilizing the term “emotion regulation” and excluding other similar concepts or grey literature to establish focus scope, e.g., trait features such as alexithymia. In running the analysis, not many variables can be used for meta-regression (i.e., gender and age), as those are the only variables included in most studies. A deep and proper understanding of the relationship between ER and PSU marked ramifications for future directions of developing approaches that are effective to curb this negative phenomenon. This paper made it very clear that there is a sizable association between these two, but it lacks support information from longitudinal and qualitative studies in detail on what kind of association ER and PSU have on each other, whether single or bidirectional, and how this association came to be. Some recommendations for future studies include going for longitudinal studies, such as prospective cohort studies or interventional studies, as well as standardizing the concept and measurement of ER or ED.

## 5. Conclusions

This systematic review and meta-analysis suggests a moderate relationship between PSU and ED, although the direction of this relationship between ED and PSU could not be ascertained. We are aware of the complexity of the subject matter, as different conceptual frameworks, definitions, and different rating scales were used to assess PSU and ER. However, the strength of our studies was based on meta-regression, which found that the young age of exposure to the smartphone is strongly associated with problems with ER and PSU.

Policymakers and health authorities should be aware of our findings for future planning to tackle both rising issues of ED and PSU, i.e., the use of strategies to improve emotion regulation, such as the use of social–emotional learning, and psychoeducation on the danger and risk of early exposure to the use of smartphones, early detection, as well as referral and management of the craving for excessive use of smartphones among our younger generations.

## Figures and Tables

**Figure 1 ijerph-19-15848-f001:**
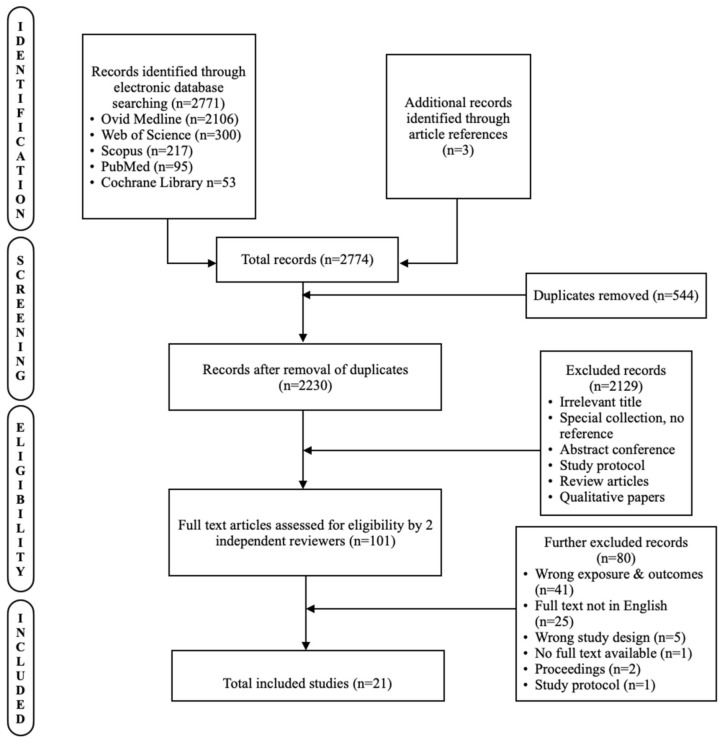
The PRISMA flowchart describing the flow of information through the different steps taken for this systematic review.

**Figure 2 ijerph-19-15848-f002:**
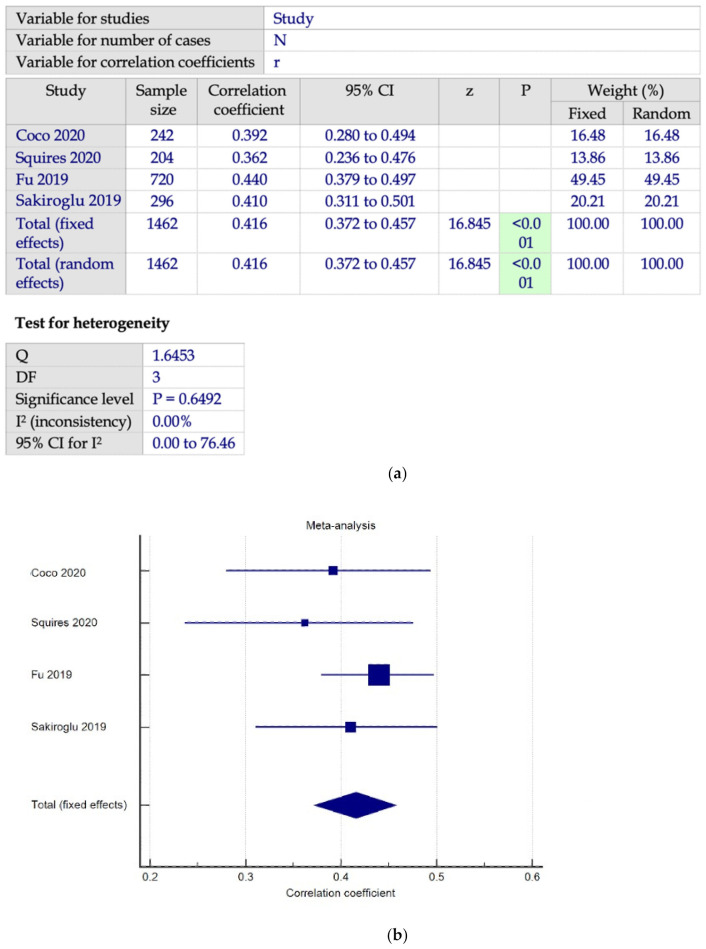
(**a**) Meta-analysis on DERS-SAS subgroup [48,51,53,65]; (**b**) forest plot showing the relationship between emotional dysregulation and PSU in studies using DERS and SAS scales [48,51,53,65].

**Figure 3 ijerph-19-15848-f003:**
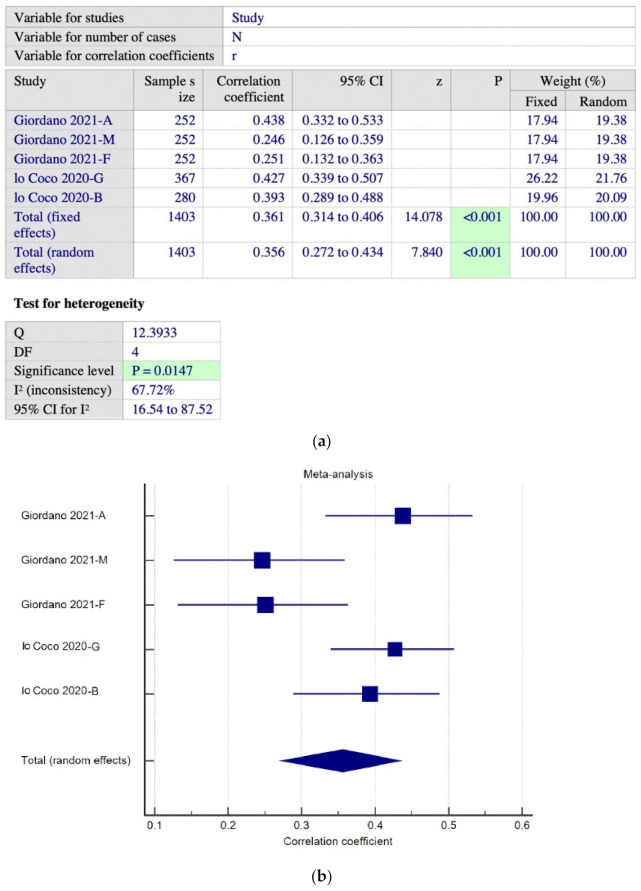
(**a**) Meta-analysis on DERS-SF—SPAI subgroup [49,50]; (**b**) forest plot showing the relationship between emotional dysregulation and PSU in studies using DERS-SF and SPAI scales [49,50].

**Figure 4 ijerph-19-15848-f004:**
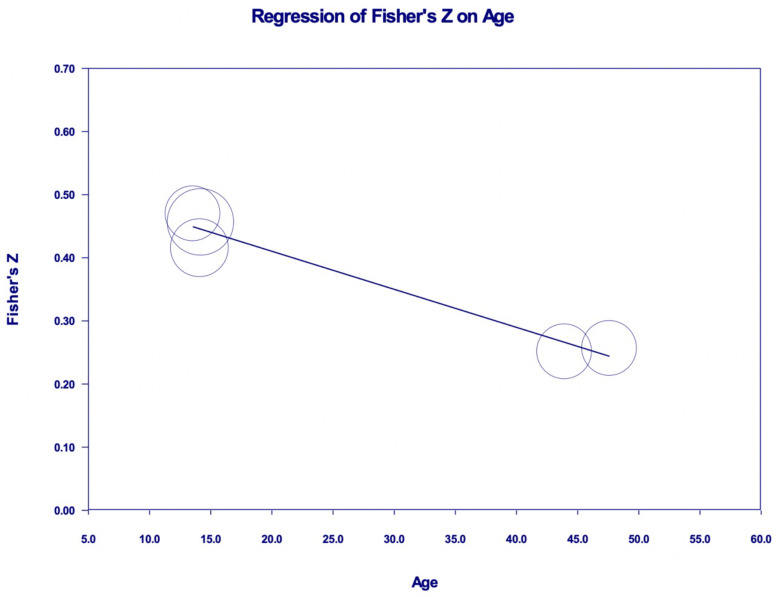
Meta-regression bubble plot showing association between subjects’ age (in the DERS-SF—SPAI subgroup) and Fisher’s Z score.

**Figure 5 ijerph-19-15848-f005:**
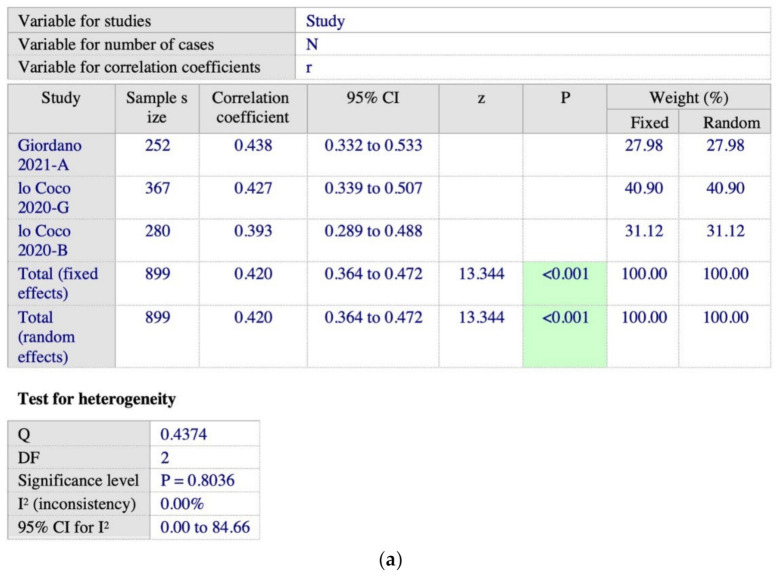
(**a**) Meta-analysis on DERS-SF—SPAI subgroup [49,50]; (**b**) forest plot showing the relationship between emotional dysregulation and PSU in studies using DERS-SF and SPAI scales after removing the 2 parents’ group [49,50].

**Figure 6 ijerph-19-15848-f006:**
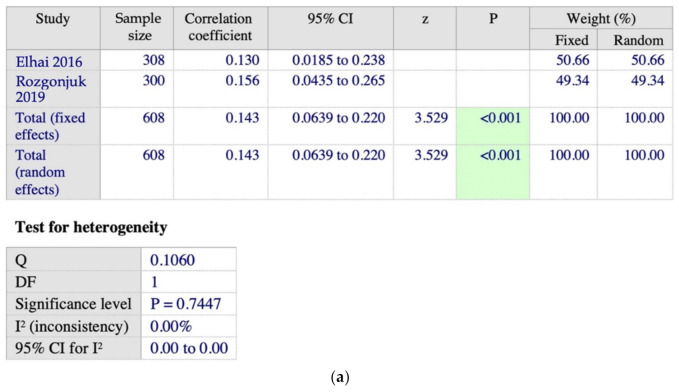
(**a**) Meta-analysis on the ERQ—SAS subgroup on the domain of “expressive suppression” [40,52]; (**b**) forest plot showing the relationship between emotional dysregulation and PSU in studies using ERQ (“expressive suppression” domain) and SAS scales [40,52].

**Table 1 ijerph-19-15848-t001:** Body evidence matrix.

Components	Grade	Comments
1. Evidence base	D—Poor	All 21 studies included are level IV studies
2. Consistency	B—Good	Out of 21 studies, 17 have similar conclusions. Only one study showed inconsistency and the rest have unique ways of communicating the results, though still relevant to the research question.
3. Clinical impact	C—Moderate	Included studies had a shared goal of establishing the association between ER and PSU. Recognizing this relationship may invoke individual insight on how they utilize their smartphones in relation to their emotional state and encourage non-hazardous individual use of the device with or without professional help.
4. Generalizability	C—Satisfactory	Studies included were carried out in different countries with subjects mainly of a younger population age group and with a formal education background. Nonetheless, the evidence can sensibly be generalized to all smartphone users.
5. Applicability	C	Body of evidence offer some support for recommendation(s), but further research is warranted to aid deeper understanding of the issue.

## Data Availability

Not applicable.

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
