# Peer review of "The Relationship between Emotion Regulation (ER) and Problematic Smartphone Use (PSU): A Systematic Review and Meta-Analyses"

_ijerph, 2022, doi:10.3390/ijerph192315848_

Round 1

Reviewer 1 Report

Comments to the Author

Summary:

The authors conducted a meta-analysis focusing on the relationship between emotion dysregulation (ED) and problematic smartphone use (PSU). Twenty-one studies were included in the systematic review. Subgroup meta-analyses confirmed a moderate positive correlation between ED and PSU. Furthermore, meta-regression analysis showed a stronger correlation between ED and PSU in younger age group.

Although overall I think the manuscript is well-written and may be of broad interest, I have a few concerns.

Major concerns:

-       In Introduction, the authors introduced the concept and measurement of emotion dysregulation and PSU separately. However, I still can't see the scientific rationale of digging out the relation between ED and PSU. A further elaboration of this part is clearly warranted.

-     The description of the statistical analysis is too brief. There is no information of the type of extracted data in meta-analysis, e.g., correlation coefficient, p-value. And the steps of how did the authors carry out the data in meta-analysis using CMA is needed.

-     In Discussion, I think the authors put too much efforts on the less important results rather than focused on the main results.

Minor concerns:

-       In the result of NHMRC, all included studies were rated as level IV. There is a lack of explanation of why giving such rating and what the meaning it is.

-       The authors ran three different subgroup analyses using the following scale combinations: DERS and SAS scales, DERS-SF and SPAI scales, ERQ and SAS. I’m wondering the rationale for choosing these combinations instead of others.

Author Response

Thank you for taking the time to review our research article. We appreciate your comments and have addressed your concerns as outlined below. Our response is typed in red font.

Major concerns:

Point 1:

In Introduction, the authors introduced the concept and measurement of emotion dysregulation and PSU separately. However, I still can't see the scientific rationale of digging out the relation between ED and PSU. A further elaboration of this part is clearly warranted.

Response 1:

We have revised the Introduction which includes inserting a 3rd paragraph that describes the most likely relation between ED and PSU. It reads as below:

"When emotion becomes dysregulated, attempts to attenuate the resulting emotional intensity could include maladaptive coping in the form of behavioral addiction; among which is problematic smartphone use (PSU).  Smartphones are highly addictive technology that has enveloped society due to their ease of use, convenience, and almost infinite access. Furthermore, the increasing importance of smartphones in our daily life these modern days blurs the line in determining whether individuals’ usage of smartphones is adaptive or maladaptive to their emotional coping. Smartphones use is a potentially maladaptive mechanism to ER as its use exposes a person to various emotionally triggering stimuli which could further dysregulate one’s emotions."

We would also like to direct your attention to paragraph 4 in the Introduction (lines 159-165, page 3/39) where we have provided some scientific evidence from the literature regarding the negative effects of PSU and mention how PSU is considered to be a form of emotion dysregulation. As this relationship is yet to be conclusive, hence the rationale of conducting this research.

Point 2:

The description of the statistical analysis is too brief. There is no information of the type of extracted data in meta-analysis, e.g., correlation coefficient, p-value. And the steps of how did the authors carry out the data in meta-analysis using CMA is needed.

Response 2:

  • We have updated the description of statistical analysis (under the Method section, lines 232-241, page 4/39 of the article) as shown below:

"Items extracted included information concerning descriptive data and statistical data for running meta-analysis. The descriptive data extracted includes (1) first author and publication year, (2) country of origin, (3) study setting, (4) sample characteristics (sample size, age group), (5) PSU measures and domains, (6) ER measures and domains, (7) the descriptive relationship between PSU’s and ER’s domain with p-value expressed in *. The statistical data for running the meta-analysis extracted includes (1) PSU measures (mean, SD), (2) ER measures (mean, SD), (3) sample size, (4) correlational coefficient together with its 95% confidence interval (CI), (5) p-value, (6) and the relevant data for running meta-regression. Relevant authors were contacted [47–53] for clarification of incomplete data, in which all but 1 author replied to get extra information required for the review and analysis."  

  • As for the type of data extracted for the meta-analyses, some changes were made to the content under subheading 2.6 Statistical analysis (pages 4-5/39), as shown below:

"A meta-analysis was performed using the Comprehensive Meta-Analysis software program (CMA) (Biostat Inc., New Jersey, USA) to determine the relationship between emotional regulation with PSU. Only studies reporting the Pearson’s correlation coefficient between ER and PSU were included in the meta-analysis.  The Pearson’s correlation coefficient together with its 95% CI was computed into CMA where the pooled effect estimates were automatically generated by the meta-analysis software. A random effect (RE) model was used for the summarized effect estimates to account for heterogeneity across studies. Analysis of data were conducted according to the different scales used to assess ER and PSU e.g. DERS, EMSS, and ERQ scale for ER versus SAS, SPAI, TMD, and APU scale for PSU. The I2 index statistic was used to describe the heterogeneity between the included studies."

  • As for the steps of how we carry out the meta-analysis of the data using CMA, the details are outlined below:

Our data was extracted into an excel sheet, with information in columns stating the name of the study, mean and SD of PSU measures, mean and SD of ER measures, sample size (N), correlation coefficient (r), and p-value.

Data was entered into the CMA program according to the accompanying user manual upon subscription of the software (Section 6: Importing data from other programs). This section has a step-by-step guide on how we can import data from an excel sheet to the CMA program.

The manual can be downloaded from Meta-Analysis Manual V3.pdf

Point 3:

In Discussion, I think the authors put too much efforts on the less important results rather than focused on the main results.

Response 3:

We have highlighted the given important results based on our meta-analysis study. Our main findings are the establishment of a moderate relationship between PSU and ED as well as the contribution of the age factor of younger smartphone users towards strengthening this relationship. We have allocated the whole of paragraphs 4 and 5 (pages 19-20/39) of the Discussion section to discuss these summarized findings. However, as the ED and PSU outcome measures are not as straightforward, different smaller constructs or terms have been used to represent them in different studies that we included. We regard these smaller findings (i.e., lack of emotional clarity, impulse control difficulties, and negative cognitive appraisal, such as catastrophizing, self-blaming, and blaming others has a positive correlation with PSU") as part of the main findings explained in alternative ways. We feel there is a need to discuss the constructs used as the original studies intended to avoid misinterpretation. We hope that by covering the different aspects of ER and PSU, our discussion would initiate interest in other researchers to explore the different aspects of this relationship that has not been clearly addressed or answered in our study results. This may be pivotal in future research to find mechanisms or theories that best explain the relationship.

Minor concerns:

Point 1: 

In the result of NHMRC, all included studies were rated as level IV. There is a lack of explanation of why giving such rating and what the meaning it is.

Response 1:

We used the NHMRC guideline as an approach to grade evidence recommendations based on the included study findings in this systematic review. Modified McMaster's instrument was used to assess the quality of the individual study. The justification of the NHMRC overall score is shown in Table 1 Body evidence matrix (page 18/39). There are scoring of 5 components to decide the overall final score. The scoring of each component is given according to the “NHMRC levels of evidence and grades for recommendations for developers of guidelines”, downloadable from the link below:

https://www.nhmrc.gov.au/sites/default/files/images/NHMRC%20Levels%20and%20Grades%20(2009).pdf

According to the guideline, in an aetiological study, level IV evidence refers to a study employing a cross-sectional or case series design. The concern of all included studies is of level IV evidence is reflected in a rating of “D” in the individual component of “evidence base”. As all other components are ranging from B to C, an overall score of “C” is given.

Point 2:

The authors ran three different subgroup analyses using the following scale combinations: DERS and SAS scales, DERS-SF and SPAI scales, ERQ and SAS. I’m wondering the rationale for choosing these combinations instead of others.

Response 2:

The meta-analyses performed used secondary data and the combinations used for the subgroup analyses were based on the available data from the included studies. Our data synthesis includes grouping the studies according to the scales used to assess the relationship between ER and PSU, hence the combinations are presented in the subgroup analysis. The rationale for performing such synthesis is to limit heterogeneity across the included studies.

After the data extraction, we ran a meta-analysis from the eligible data set and found a pool correlation coefficient, r= 0.286 (N=3793, 95%CI 0.160-0.402, p<0.0001, I2=94.03%). However, in view of the high heterogeneity (I2=94.03%) across studies, it deems not appropriate to apply a meta-analysis of correlation [Hafdahl et al 2009, Field 2005].

We later proceed with subgroup analysis as this is a more appropriate way considering that the high heterogeneity can successfully be removed.

The three scales combination (DERS and SAS, DERS-SF and SPAI, ERQ and SAS) are used according to the availability of data. The included studies applied these scale combinations.

Reviewer 2 Report

The article is very interesting and addresses a topic of relevant interest, also concerning educational psychology. 

To make the reading more agile, I suggest reorganising the structure of Tables 1,2,3,4 better. Include Table 1 as supplementary material, and Tables 2,3,4 as an appendix. 

With regard to the conclusions, it would be opportune to expand them by highlighting more the effects of the results in the field of psychological and educational interventions.

Author Response

Thank you for taking the time to review our research paper. We appreciate your comments and concerns. We have addressed your concerns as outlined below. Our responses are typed in red font.

Point 1:

To make the reading more agile, I suggest reorganising the structure of Tables 1,2,3,4 better. Include Table 1 as supplementary material, and Tables 2,3,4 as an appendix. 

Response 1:

There were changes applied to the table structure as suggested. Table 1 has been removed from the manuscript, renamed as Table S1, and included as supplementary material (page 6/39). Tables 2, 3, and 4 has been renamed as Tables A1, A2, and A3, and included in the Appendix (page 8/39, and 11/39). Due to the above changes, Table 5 was renamed to Table 1 (page 18/39).

Point 2:

With regard to the conclusions, it would be opportune to expand them by highlighting more the effects of the results in the field of psychological and educational interventions.

Response 2:

Thank you for your comments. Under the conclusion subheading, we expanded our conclusion and wrote (lines 759-764, on page 22/39):

“Policymakers and health authorities should be aware of our findings for future planning to tackle both rising issues of ED and PSU, i.e., the use of strategies to improve emotion regulation such as the use of social-emotional learning, and psychoeducation on the danger and risk of early exposure to the use of smartphones, early detection, referral and management of the craving for excessive use of smartphones among our younger generations.”

Reviewer 3 Report

a very interesting article and one that certainly suggests more study.   A lot of the domains cited may be very subjective - for example cell phone addiction and would be more useful if defined better - hours of use, circumstances of use

I also think that it is not clear if the cell phones cause the mental health issues, or adolescents with mental health issues use the smart phone in a problematic way and this could be mentioned in the conclusion

Author Response

Thank you for taking the time to review our research paper. We appreciate your comments and concerns and have addressed them as outlined below. Our responses are in red font.

Point 1:

A lot of the domains cited may be very subjective - for example cell phone addiction and would be more useful if defined better - hours of use, circumstances of use

Response 1:

We agree that the concept of “cell phone or smartphone addiction” has not been standardized, that was why we brought up this issue in our Introduction.

  • We chose the term PSU as it is often used by researchers and covers most of the aspects of problematic use of smartphones. This is highlighted in our Introduction (line 166-175, page 3/39):

"In measuring PSU, the most efficient or gold standard scales are yet to be established; disagreements exist on how to approach this problem [37]. Existing literature tends to measure this construct in the context of the “addiction” framework following their conceptual similarity, thus the term “Smartphone addiction” was commonly seen to be used interchangeably with PSU [38]. Smartphone usage is also gauged in the background of associated motivations, attitudes, and frequency which may reflect the mechanism behind PSU [39–41]. A relatively new term that sometimes is used interchangeably with PSU is nomophobia, derived from the word "no mobile phone phobia" which refers to the anxiety and intense discomfort of not being in touch with a mobile phone and its associated features [42,43]."

  • We also highlighted in the results part that no standardized term has been used (line 364-366, page 9/39):

"Terms such as "problematic smartphone use", "smartphone addiction", "smartphone overuse", "smartphone amount use", "smartphone use frequency", and "nomophobia" were summarized according to findings as “problematic smartphone use”, abbreviated as "PSU"."

By looking into the studies, one study (Elhai 2018) gives sub-domains of PSU for “overuse”, “withdrawal” and “tolerance”, while another study (Elhai 2018b) gives a specific division into “objective minutes of smartphone use”, one study (Fortes 2020) specifies “amount use” and another study (Hoffner 2015) specify “missed uses” of mobile phone. Overall all included studies give the concepts that we can summarize into “problematic smartphone use” which links to emotional problems.

We highlighted the non-standardized terms for problematic smartphone use, hoping to give insight into primary studies in the future to standardize the use of the term “problematic smartphone use” (PSU).

Point 2:

I also think that it is not clear if the cell phones cause the mental health issues, or adolescents with mental health issues use the smart phone in a problematic way and this could be mentioned in the conclusion.

Response 2:

Based on our reading and understanding, in meta-regression, the explanatory variables are characteristics of studies that might influence the size of the intervention effect. In our findings, there was a strong association between the age of exposure to smartphone use and PSU. According to Higgins JPT, Green S (editors) in their Cochrane Handbook for Systematic Reviews of Interventions Version 5.1. 0 [updated March 2011, The Cochrane Collaboration, 2011], they explained the meta-regressions outcome variable is predicted according to the values of one or more explanatory variables, in which the outcome variable is the effect estimate (for example, a mean difference, a risk difference, a log odds ratio, or a log risk ratio). Based on this, under the conclusion, we wrote on page 22/39:

"Policymakers and health authorities should be aware of our findings for future planning to tackle both rising issues of ER and PSU, i.e., the use of strategies to improve emotion regulation such as the use of social-emotional learning, and psychoeducation on the danger and risk of early exposure to the use of smartphones, early detection as well as referral and management of the craving for excessive use of smartphones among our younger generations."

However, despite the association clearly being present between these 2 study variables, we do agree that our findings could not imply a causality effect on whether ED causing PSU or PSU is the one causing ED. Hence, we have made some changes to the conclusion based on your suggestion, as shown below:

"This systematic review and meta-analysis suggest a moderate relationship between PSU and ED although the direction of this relationship between ED and PSU could not be ascertained. We are aware of the complexity of the subject matter as it has different conceptual frameworks, definitions and different rating scales were used to assess PSU and ER."

Round 2

Reviewer 1 Report

There is no other suggestions. The new version is greater improved than the version one.